# Contextual Factors in Ethnic-Racial Socialization in White Families in the United States

**Tanya Nieri** * and **Justin Huft**

Department of Sociology, University of California at Riverside, Riverside, CA 92521, USA; jhuft001@ucr.edu
* Correspondence: tanyan@ucr.edu

**Abstract:** Recent demographic shifts and sociopolitical events in the United States have led to a racial reckoning in which white people are engaging with issues of race and racism in new ways. This study addressed the need for research to better understand contextual factors in ethnic-racial socialization (ERS)—strategies in white families to teach children about their own and other people's ethnicity or race. It examined the relation of neighborhood, school, and social network ethnic-racial composition and U.S. region of residence to participants' perceptions of ethnic-racial socialization by parents. It employed a large, national survey sample of white young adults reporting on their ERS while growing up and a comprehensive set of ERS strategies. We found that the ethnic-racial composition of the family's social network, but not the neighborhood or school, was related to exposure to ERS: the whiter the network, the less frequent the socialization, particularly antiracism socialization and exposure to diversity. We also found that Southern residents were more likely than residents in the West and Midwest to be exposed to the strategies of preparation for bias, mainstream socialization, and silent racial socialization. The findings show that these two contextual factors relate to both the frequency and content of the ERS a white child receives.

**Keywords:** ethnic-racial socialization; racial socialization; whites; ethnic-racial composition; geographic region; region of residence

## 1. Introduction

Changing U.S. demographics and recent sociopolitical events have changed the discourse on race in America. The U.S. is more ethnically and racially diverse than in the past, and white children are more likely than in the past to interact more frequently with children of color [1,2]. National protests and the emergence of the Black Lives Matter movement in response to racialized police violence as well as the 2016 US presidential election of Donald Trump, who openly supported white supremacist causes and engaged in racist rhetoric, have raised the salience of race and racism as issues that Americans must face [3]. White folks are engaging in new ways with these issues, and there is a need to understand what conversations are happening in white families and what messages children receive [2,3].

At the same time, researchers focusing on parents' ethnic-racial socialization (ERS) of their children have come to recognize the need to expand their focus beyond families of color to include white families [4,5]. Parents' socialization of their children is one avenue by which racial ideologies may be reproduced and in turn, reinforce structures of racial oppression [6,7]. This paper aims to expand the understanding of ERS by white parents of their white children by examining exposure to ERS, using a national U.S. sample of white young adults reporting on their parents' ERS while growing up. Specifically, we ask how the ethnic-racial composition of the neighborhood, the school, and the family social network, and the geographic region of residence relate to young adults' perceptions of ERS by their parents.

### 1.1. Ethnic-Racial Socialization (ERS)

ERS refers to parents' strategies to teach their children about their own and other people's ethnicity or race [4,8]. It may involve explicit strategies, such as messages conveyed through conversations about ethnicity-race, or implicit strategies, such as messages conveyed through choices about where to live and what schools to attend [2]. The most commonly studied strategies are cultural socialization (messages about the family's ethnic-racial heritage and traditions), promotion of mistrust (messages encouraging wariness of other ethnic-racial groups), and preparation for bias (messages about the possibility of being the victim of ethnic-racial discrimination) [5]. Other strategies include egalitarianism (messages about the equality of ethnic-racial groups) and mainstream socialization (messages de-emphasizing ethnicity-race and endorsing mainstream (white) cultural institutions and values, such as individualism) [4,5]. Recent research on white families has explored additional strategies: antiracism socialization (messages about white privilege and structural racism) [2,9–11], color-blind socialization (messages deemphasizing the importance of race or discouraging discussion of race) [12–15], and color-conscious socialization (messages about the value of ethnic-racial diversity and actions to expose children to diversity) [2,15,16].

### 1.2. Contexts of Ethnic-Racial Socialization: Children Learning and Parents Teaching

Prior research on ERS has documented how parents' characteristics influence whether and how parents engage in ERS [4,5]. Yet, little research has examined how contextual factors affect the frequency and content of parents' engagement in ERS. Some research on ERS has examined the relation of contextual factors, including neighborhood, school, and social network, to children's socialization outcomes. For example, prior research has shown that exposure to neighborhood diversity shapes children's racial attitudes [17]. Similarly, schools are socializing contexts; they provide inter-group contact, information on ethnicity-race, and opportunities for processing ethnic-racial messages from other contexts [18]. Ethnically-racially diverse schools, in particular, may help students to maintain more nuanced understandings of ethnicity-race, better navigate ethnicity-race in other contexts, and develop among white students a white identity that involves greater recognition of other ethnic-racial groups [4]. Inter-group contact in peer social networks also shapes racial attitudes. For example, inter-racial friendship and dating predict a tendency to care about racial equity [19].

Parents' messages to children about ethnicity-race play a role in helping children process their experiences in various contexts. Just as children are influenced by their contexts, parents are influenced by their contexts. Thus, it is important to understand how certain contextual factors relate to exposure to specific ethnic-racial socialization messages from parents. In this paper we explore how neighborhood, high school, and family social network composition and geographic region relate to the frequency and content of socialization by parents.

### 1.3. Ethnic-Racial Composition in the Neighborhood, School, and Social Network

Contact theory, strongly supported by empirical research, argues that greater inter-group contact reduces prejudice [20]. We expect that greater diversity in ethnic-racial composition will be associated with greater exposure to ERS, because parents exposed to greater diversity in the neighborhood, school, and family social network are likely to view ethnicity-race as more salient and, in turn, seek to socialize their children about it. Recent quantitative research supports this reasoning, showing that neighborhood, school, and social network ethnic-racial composition relate to the extent and nature of ERS in white families. For example, Eveland and Nathanson [21] found that white parents in highly white counties discussed racism with their children less frequently than white parents in racially diverse counties. Zucker and Patterson [22] found that white parents whose children attended more ethnically-racially diverse schools reported more frequent preparation for bias. Barner [23] and Perry and colleagues [24] found that white parents'

interracial contact was positively related to engagement in ERS, particularly egalitarianism and acknowledgement of racism.

Recent qualitative research also highlights ways in which the neighborhood, school, and family social network relate to exposure to ERS in white families. Hagerman [2] found that among affluent white families, parents' choices to live in racially homogeneous neighborhoods and send their children to racially homogeneous schools were related to parents' racialized perceptions about the kinds of people in those places and, in turn, sent implicit color-blind socialization messages to children. Parents who chose to live in diverse neighborhoods and/or send their children to diverse schools pursued more color-conscious socialization. Hagerman [2] also found that white parents with racially homogenous social networks did not view race as salient to their family and, in turn, felt less need to discuss racial issues with their children. Underhill [25] found that middle-class white parents, whether in majority-white or multiracial neighborhoods, cultivated symbolic and spatial distance between their children and their poor neighbors; they viewed poor white neighbors with disgust and, thus, avoided contact, whereas they viewed contact with black neighbors as both valuable and threatening. This research suggests that exposure to greater diversity may be associated with the content of ERS, not merely its frequency. Greater diversity in neighborhoods, schools, and family social networks may be related to greater exposure to antiracism socialization and/or color-conscious socialization and less of other ERS strategies in white families.

### 1.4. Geographic Region

Regional cultures exist and are related to attitudes and behavior [26]. The U.S. South is characterized as having its own regional culture [27], one that is related to ethnic-racial attitudes. Prior research has shown that residence in the South is associated with less awareness of Blacks' structural disadvantages [11,28] and less acceptance or tolerance of Black Americans [29]. It also shows that Southern parents are more likely to have an authoritarian parenting style, characterized by high levels of demandingness and low levels of responsiveness [30]. Given that parents' racial attitudes and parenting style are associated with ERS [11,21], parents in the South may expose their children to ERS differently than parents in other regions. Specifically, Southern parents may be less likely to respond to children's inquiries about race, and white Southern parents may be especially less likely to respond to such inquiries, given that white parents have been shown to experience race as low in salience and express racial apathy [31–35]. Therefore, white parents in the South may engage in less frequent socialization, viewing race as less salient to their families. An alternative possibility is that they may engage in a pattern of socialization that includes more frequent promotion of mistrust and less frequent antiracism socialization, given the proclivity to negative racial views associated with the region. We identified no prior studies that examined U.S. regional differences in ERS among white families.

### 1.5. This Study

Prior studies of ERS typically assess only one or a few socialization strategies. Furthermore, with the exception of Thompson [11], and Eveland and Nathanson [21], the prior research on the relation of contextual factors to ERS in white families relied on relatively small, local samples. There is a need for research that examines a comprehensive set of strategies in a single study and employs a large, national sample. Using a large, national sample and a complete set of ERS strategies, this study examines the relation of neighborhood, high school, and family social network ethnic-racial composition and geographic region to white young adults' perceived exposure to ERS while growing up.

## 2. Materials and Methods

### 2.1. Data

The data were obtained from an online survey administered to a U.S. national sample, balanced on gender and geographic region, in January 2022. We contracted with Qualtrics

to recruit participants and administer our survey. Qualtrics recruits respondents through partnering market research organizations and screens them for eligibility. Recruitment invitations are sent via email, SMS (i.e., text) notifications, Qualtrics' panel portal, or in-app notifications. Respondents are compensated for participation. During survey administration, Qualtrics screens the data, eliminates cases with flawed responses (e.g., response sets), and replaces deleted cases with new ones until the desired sample size is reached. Upon completion of data collection, Qualtrics provided us with an anonymized copy of the data. We note that the study occurred in the middle of the global COVID-19 pandemic.

*2.2. Sample*

Although 1009 people completed the survey, we retained a sample of 988 for this analysis, excluding 11 people with incomplete data. Eligible participants were emerging adults (18 to 25 years old), self-identified as white and not as bi- or multiracial, resided in a U.S. state, were born in the United States, were raised by white parents, and were not adopted. We focused on young adults, a strategy employed by many other ERS studies [8] to avoid the pitfalls associated with parent reports—in particular, sensitivity to social desirability and interviewer or respondent bias (e.g., fear of being viewed as racist or prejudiced) [4]. Independent samples' *t*-tests revealed no statically significant differences between the excluded and included groups on ERS and ethnic-racial composition. Similarly, chi-square tests revealed no statically significant associations between sample inclusion/exclusion and region of residence. Sample descriptives are contained in Table 1.

**Table 1.** Sample descriptives.

|  | Mean (SD) |
|---|---|
| Cultural socialization | 2.70 (1.02) |
| Promotion of mistrust | 2.03 (1.15) |
| Preparation for bias | 2.45 (1.03) |
| Egalitarianism | 2.59 (0.96) |
| Mainstream socialization | 2.63 (1.01) |
| Exposure to diversity | 2.64 (1.12) |
| Silent racial socialization | 1.92 (1.00) |
| Antiracism socialization | 2.50 (1.16) |
| % white in neighborhood | 61.12 (27.75) |
| % white in high school | 58.46 (25.91) |
| % white in family's social network | 64.31 (27.2) |
|  | **Percentage** |
| Region of residence |  |
|     Northeast | 17.0 |
|     Midwest | 21.2 |
|     West | 24.9 |
|     South | 36.9 |
| Gender |  |
|     Man | 44.6 |
|     Woman | 53.8 |
|     Non-binary/other gender | 1.50 |
| Parents' education |  |
|     Less than eighth grade | 0.20 |
|     Eighth grade | 1.60 |
|     High school/GED | 32.9 |
|     Vocational/associate's | 17.5 |
|     Bachelor's | 29.5 |
|     Advanced degree | 18.3 |

Using available race-by-age data on education, we compared our sample and the national population of white young adults and found their levels of completed education to be similar. The US Department of Commerce [36] reported that among white people 18 to 24 years old, 11.8% had less than a high school education, 28.5% completed high school, 48.2% had an associate's degree or some college, and 11.5% had a bachelor's or higher degree. In our sample, which includes 25-year-olds in addition to 18–24-year-olds, 2.7% completed less than high school, 37.8% completed high school or GED, 46.4% completed vocational education, an associate's degree, or some college, and 13.1% complete a bachelor's or higher degree.

*2.3. Measures*

The ERS measures capture respondents' perceptions of socialization during childhood. Consistent with other ERS measures for young adults [37], they ask respondents to report on ERS "when you were growing up". Each measure indicates the perceived average frequency of component socialization items. The cultural socialization (5 items, α = 0.877) (e.g., "How often did your parent[s] teach you about important people or events in the history of your racial/ethnic group?"), promotion of mistrust (3 items, α = 0.909) (e.g., "How often did your parent[s] do or say things to keep you from trusting members from other racial/ethnic groups?"), and preparation for bias (5 items, α = 0.867) (e.g., "How often did your parent[s] talk with you about stereotypes, prejudice, and/or discrimination against people of your racial/ethnic group?") measures were modified versions of Tran and Lee's measures [38,39]. The egalitarianism measure (6 items, α = 0.811) (e.g., "How often did your parent[s] tell you that all races are considered equal?") was a modified version of the contrast egalitarianism scale of the M-Racial Bias Preparation Scale from Langrehr, Thomas, and Morgan [40]. We created the mainstream socialization measure (4 items, α = 0.759) (e.g., "How often did your parent[s] tell you that a person's individual characteristics are more important than the characteristics of the group(s) to which they belong?"), drawing on Rollins [41]. Silent racial socialization (α = 0.889) had 5 items: " . . . dismiss your experience of race", " . . . discourage conversations about race in the United States", " . . . discourage you from exploring your racial heritage", " . . . avoid discussing their own racial experiences with you", and " . . . tell you to avoid talking about race with other people" [42]. Informed by the qualitative literature which has examined these constructs [2,13,14,33,35], we created measures for exposure to diversity (4 items, α = 0.864) (e.g., "How often did your parent[s] encourage you to be friends with people from other racial-ethnic groups?") and antiracism socialization (3 items, α = 0.846) (e.g., "How often did your parent[s] speak with you about how white people have an advantage in life because of their race?"). In the case of existing measures, some modification was necessary to make them comparable to each other in the present study. All measures, thus, have a uniform preamble ("When you were growing up. . . ") and response set (1 = never, 2 = rarely, 3 = sometimes, 4 = often, 5 = very often). Each measure's Cronbach's alpha was greater than 0.70, indicating good reliability.

Contextual variables included ethnic-racial composition and region of residence. All three composition measures were based on single items, each with a sliding scale from 0–100. The low end of the scale represented 0% same ethnicity-race composition and the high end represented 100% same ethnicity-race composition. For neighborhood composition, we asked, "Looking back on the neighborhood where you lived during high school (if you lived in more than one neighborhood, think of the neighborhood where you lived the longest), what percentage of residents were members of your racial-ethnic group?" For high school composition we asked, "What percentage of students at your high school were members of your racial-ethnic group?" For the family's social network composition, we asked, "When you were growing up, what percentage of your family's friends and acquaintances were members of your racial-ethnic group?" We employed dummy variables for participants' region of residence ("Northeast", "Midwest", "West") with "South" as the reference group.

As covariates, we included participants' gender and parents' highest level of education completed, given prior research documenting their relation to ERS [8]. We included dummy variables for "Woman" and "Non-binary or other gender", with "Woman" as the reference group. The measure for parents' highest level of education was continuous, ranging from 1 "Less than 8th grade" to 6 "Master's degree or higher".

*2.4. Analysis*

We produced descriptive statistics on all measures. We examined correlations to assess for bivariate relations between ethnic-racial composition and ERS. We conducted analysis of variance (ANOVA) with post hoc Bonferroni tests to examine mean differences in ERS by region. We conducted ordinary least squares regression to assess for multivariate relations; our set of models regressed each ERS measure on the contextual variables and covariates. We assessed for and found no collinearity, with all variance inflation factors $\leq 2$. We report as statistically significant results with probability values of $\leq 0.05$.

## 3. Results

As indicated in Table 1, the means of the frequency of ERS were low, indicating that parents engaged in ERS, on average, only sometimes or rarely while participants were growing up. The highest mean frequency was for cultural socialization. The lowest mean frequency was for silent racial socialization. Regarding ethnic-racial composition, participants lived during high school in a neighborhood that, on average, was 62% white, attended a high school that was, on average, 58% white, and had friends and acquaintances while growing up who were 64% white, on average. Regarding region of residence, the largest group of participants resided in the South (36.9%), followed by the West (24.9%), the Midwest (21.2%), and the Northeast (17%).

Table 2 shows the correlations of ERS with ethnic-racial composition. With one exception, the correlations were negative. In the case of neighborhood composition and high school composition, the correlation was only statistically significant in the case of promotion of mistrust and preparation for bias; the higher percentage white the neighborhood and high school, the less frequently parents engaged in promotion of mistrust and preparation for bias. In the case of the family's social network composition, the correlation was statistically significant in the case of promotion of mistrust, preparation for bias, exposure to diversity, silent racial socialization, and antiracism socialization; the higher the percentage of white friends and acquaintances, the less frequent was the socialization.

**Table 2.** Pearson correlations of perceived ERS with ethnic-racial composition.

|  | % White Neighborhood | % White High School | % White Family Social Network |
|---|---|---|---|
| Cultural socialization | −0.056 | −0.051 | −0.062 |
| Promotion of mistrust | −0.119 ** | −0.113 ** | −0.075 * |
| Preparation for bias | −0.090 ** | −0.088 ** | −0.100 ** |
| Egalitarianism | −0.048 | −0.026 | −0.055 |
| Mainstream socialization | −0.007 | 0.033 | −0.006 |
| Exposure to diversity | −0.043 | −0.045 | −0.082 * |
| Silent racial socialization | −0.055 | −0.053 | −0.073 * |
| Antiracism socialization | −0.009 | −0.034 | −0.101 ** |

Note. * $p < 0.05$ (2-tailed). ** $p < 0.01$ (2-tailed).

Table 3 shows the mean frequencies of ERS by region of residence. ANOVA tests revealed statistically significant differences in the means for promotion of mistrust (F = 2.848, df = 987, $p = 0.037$), egalitarianism (F = 2.816, df = 987, $p = 0.038$), mainstream socialization (F = 5.894, df = 987, $p = 0.001$), and silent racial socialization (F = 5.274, df = 987, $p = 0.001$). Post hoc Bonferroni tests revealed that participants who resided in the West were different from participants who resided in the South; they reported lower mean frequencies of promotion of mistrust ($p = 0.034$), egalitarianism ($p = 0.026$), mainstream socialization

($p$ = 0.001), and silent racial socialization ($p$ = 0.001). They were also different from participants who resided in the Midwest; they reported a lower mean frequency of silent socialization ($p$ = 0.034).

**Table 3.** Means and standard deviations of perceived ERS by region of residence.

| | Northeast | Midwest | West | South |
|---|---|---|---|---|
| Cultural socialization | 2.75 (2.1) | 2.62 (1.01) | 2.65 (1.06) | 2.76 (1.0) |
| Promotion of mistrust | 2.10 (1.16) | 2.00 (1.14) | 1.87 (1.14) * | 2.13 (1.15) |
| Preparation for bias | 2.49 (0.93) | 2.30 (1.03) | 2.41 (1.09) | 2.53 (1.02) |
| Egalitarianism | 2.56 (0.89) | 2.57 (1.02) | 2.46 (0.94) * | 2.69 (0.97) |
| Mainstream socialization | 2.70 (1.01) | 2.55 (1.01) | 2.44 (0.98) *** | 2.77 (1.02) |
| Silent racial socialization | 1.96 (0.98) | 1.97 (1.04) ** | 1.71 (0.94) *** | 2.02 (1.01) |
| Exposure to diversity | 2.63 (1.10) | 2.49 (1.13) | 2.72 (1.18) | 2.68 (1.09) |
| Antiracism socialization | 2.58 (1.11) | 2.42 (1.15) | 2.52 (1.19) | 2.50 (1.17) |

Note. * Statistically significantly different from South at the $p < 0.05$ level. ** Statistically significantly different from West at the $p < 0.05$ level. *** Statistically significantly different from South at the $p < 0.001$ level.

The multivariate results are presented in Table 4. Regarding ethnic-racial composition, only family social network composition was statistically significantly related to ERS, controlling for covariates. The greater the percentage white of the social network, the less frequent was the reported exposure to diversity and antiracism socialization. In the case of region of residence, there was a pattern of negative coefficients, but only three estimates were statistically significant. Being from the Midwest and West, relative to being from the South, was statistically significantly related to ERS, controlling for covariates. Specifically, participants from the Midwest reported less frequent preparation for bias and mainstream socialization than participants from the South, and participants from the West reported less frequent mainstream socialization and silent racial socialization than participants from the South.

**Table 4.** Standardized estimates from regressions of perceived ERS on contextual factors and covariates.

| | Cultural Socialization | Promotion of Mistrust | Preparation for Bias | Egalitarianism | Mainstream Socialization | Silent Racial Socialization | Exposure to Diversity | Antiracism Socialization |
|---|---|---|---|---|---|---|---|---|
| | B (SE B) | B (SE B) | B (SE B) | B (SE B) | B (SE B) | B (SE B) | B (SE B) | B (SE B) |
| % white ethnic-racial composition | | | | | | | | |
| % neighborhood | −0.001 (0.002) | −0.003 + (0.002) | −0.001 (0.002) | −0.001 (0.002) | −0.002 (0.002) | 0 (0.002) | 0 (0.002) | 0.004 + (0.002) |
| % high school | −0.001 (0.002) | −0.003 (0.002) | −0.001 (0.002) | 0.001 (0.002) | 0.003 + (0.002) | 0 (0.002) | 0 (0.002) | 0 (0.002) |
| % family social network | −0.001 (0.002) | 0.001 (0.002) | −0.002 (0.002) | −0.002 (0.001) | 0.001 (0.002) | −0.002 (0.002) | −0.004 * (0.002) | −0.007 *** (0.002) |
| Region † | | | | | | | | |
| Northeast | 0.021 (0.096) | −0.001 (0.108) | 0.01 (0.096) | −0.083 (0.090) | −0.014 (0.095) | −0.021 (0.094) | −0.048 (0.106) | 0.086 (0.109) |
| Midwest | −0.122 (0.088) | −0.108 + (0.099) | −0.18 * (0.089) | −0.09 (0.084) | −0.194 * (0.087) | −0.031 (0.087) | −0.182 + (0.098) | −0.076 (0.101) |
| West | −0.046 (0.089) | −0.183 + (0.1) | −0.015 (0.089) | −0.131 (0.084) | −0.222 * (0.088) | −0.222 * (0.087) | 0.046 (0.098) | 0.029 (0.101) |
| Covariates †† | | | | | | | | |
| Man | 0.093 (0.069) | 0.12 (0.078) | 0.176 * (0.069) | 0.202 ** (0.065) | 0.237 *** (0.068) | 0.167 * (0.068) | −0.032 (0.076) | 0.04 (0.079) |
| Non-binary or other gender | −0.873 *** (0.266) | −0.293 * (0.299) | −0.557 * (0.267) | −0.034 (0.251) | −0.144 (0.262) | −0.31 (0.26) | −0.433 (0.294) | 0.083 (0.303) |
| Parents' education | 0.06 * (0.028) | −0.096 ** (0.031) | 0.027 + (0.028) | −0.011 (0.026) | 0.029 (0.028) | 0.037 (0.027) | 0.082 ** (0.031) | 0.091 ** (0.032) |
| Model fit | | | | | | | | |
| Adjusted $r^2$ | 0.017 | 0.028 | 0.021 | 0.013 | 0.025 | 0.021 | 0.013 | 0.017 |
| N of cases | 988 | 988 | 988 | 988 | 988 | 988 | 988 | 988 |

Note. † South is the reference group. †† Woman is the reference group. + $p < 0.10$, * $p < 0.05$, ** $p < 0.01$, *** $p < 0.001$.

Being a man, relative to being a woman, was positively related to preparation for bias, egalitarianism, mainstream socialization, and silent racial socialization. Being non-binary or other gender was negatively related to cultural socialization, promotion of mistrust, and preparation for bias. Greater parental education was positively related to cultural socialization, exposure to diversity, and antiracism socialization and negatively related to promotion of mistrust. The adjusted $R^2$s of the models were low, indicating that the variables in the models explained only a small portion of the variance in exposure to ERS.

## 4. Discussion

This study employed data from a national sample, with a comprehensive set of ERS measures, to examine contextual factors related to white young adults' perceptions of their exposure to ERS. We found that the sample had a high degree of homogeneity in terms of ethnic-racial composition; the neighborhood, high school, and social contexts were majority white. Despite homogeneity across the three spheres, only the composition of the family's social network of friends and acquaintances, which had the most homogeneous composition, was related to young adults' perceived exposure to ERS. The null findings for neighborhood and school composition are not consistent with prior research [2,21,22]. The difference in result may be due to the fact that prior studies relied on objective data (e.g., census data and official school data) whereas our study employed self-reports which may be biased due to social desirability or other factors [35].

Our finding of a significant effect of family social network composition—less diverse composition was associated with less exposure to ERS—is consistent with prior research demonstrating the influence of ethnicity-race in social interactions and relationships [2,23,24], although there were some distinctions. While Barner [23] found that white parents' greater exposure to diverse people was related to more frequent engagement in egalitarianism, we found no relation of family social network composition to egalitarianism. This result may be due to a difference in measures: Barner [23] measured parents' reports of their social network whereas we measured young adults' reports of the family's social network. Alternatively, it may be due to the peculiarities of Barner's sample, which included only 93 participants.

We note that, in our study, family social network composition was only related to two ERS strategies—exposure to diversity and antiracism socialization—which were not measured in prior quantitative research assessing ethnic-racial composition. Thus, family social network composition relates not only to whether children receive ERS but also the specific content of their ERS. More diverse family social networks may motivate both engagement in ERS at all and engagement in specific ERS strategies. Prior qualitative research has shown that white parents who employ these two ERS strategies typically embrace ethnic-racial diversity and aim to socialize children to appreciate diversity and avoid racism [2,16,33,43]. While that research has documented problems with these strategies in terms of whether they actually yield the desired results in white children, it shows that less diversity inhibits engagement due to a perceived lack of salience of racial issues [2,23,31–35]. Our quantitative results support the prior qualitative findings that whites' social segregation from other racial groups undermines their understanding of racial issues and in turn, their willingness to socialize their children about them [16,33,44].

The results for region of residence support the hypothesis that participants from the South would be distinct from other participants in terms of ERS. The Southern participants did not receive less frequent ERS; rather, they received more of specific ERS strategies—in particular, preparation for bias, mainstream socialization, and silent racial socialization. Preparation for bias is a strategy typically employed in the context of discrimination—that is, parents engage in preparation for bias to help their children be resilient to ethnic-racial discrimination [45]. Whites, relative to other ethnic-racial groups, enjoy broad advantages on nearly all indicators (e.g., income/wealth [46], education [47], health [48], criminal justice [49]); thus, they have less reason to be concerned about bias since their children's outcomes are unlikely to be significantly affected. It may be that the higher prevalence

of negative racial attitudes in the South [11,28,29] is accompanied by greater perceived discrimination among whites and, in turn, motivates white parents in the South to engage in more frequent preparation for bias than parents elsewhere. As for mainstream socialization and silent racial socialization, these strategies downplay the significance of race and racism and, thus, are consistent with the negative racial attitudes that are more prevalent in the South. Thus, as with preparation for bias, relative to white parents in other regions, white parents in the South may be more motivated to employ these ERS strategies and view other ERS strategies as unnecessary or inconsistent with their racial views.

We note that the contextual factors were not related to cultural socialization, the ERS strategy perceived to be received most frequently. In contrast to the other ERS strategies, which focus on intergroup relations, cultural socialization focuses exclusively on the family's own ethnic-racial group. This may explain why the contextual factors, which capture information on multiple groups, were unrelated to this strategy.

Our findings show that how children are socialized about ethnicity-race is related to the social context in which the family lives. They suggest that to understand parents' role in shaping their children's understandings of ethnicity-race, we need to attend to how a family's social network—with whom they associate socially—shapes parents' understandings of racial issues, their motivation to teach children about those issues, and the specific lessons they wish to communicate. That said, although we found some relations between some contextual factors and ERS, our models had low explained variance. This result may indicate that other unmeasured contextual variables, such as the neighborhood level of discrimination [50,51], better account for the frequency and nature of ERS. Alternatively, it may mean that microsystem variables (i.e., parent and child characteristics), which have been extensively studied to date [8], have a stronger association with ERS than contextual factors.

## 5. Limitations and Future Research

Several limitations are important to consider when interpreting the results. The study was cross-sectional; therefore, we cannot definitively conclude that there were causal relations between the contextual factors and ERS. We focused on parental socialization and examined neither other sources of socialization nor children's processing of their socialization. Our region of residence variable is based on current residence, which may not be the same as where the participant resided while growing up and being socialized. Our ERS measures rely on recall; though much of the research on ERS relies on recall, an ideal study would be prospective, following people from childhood into emerging adulthood.

Future research should continue to assess the role of social and geographical contextual factors in ERS in white families. It could examine the interactive relations between the context and different agents of socialization [8] and between the context and children's responses to socialization [2]. It could expand to include assessment of other contextual variables (e.g., urban versus rural, economic segregation, media representations, political context). Method-wise, scholars should seek to employ longitudinal designs, use official composition data rather than self-reports, and obtain data on childhood region of residence. Finally, future research could examine whether and how the COVID-19 pandemic might have altered, for the next cohort of young adults, the relations examined here, given that the pandemic disrupted communities and families.

**Author Contributions:** Conceptualization, T.N.; methodology, T.N.; formal analysis, J.H. and T.N.; writing—original draft preparation, T.N.; writing—review and editing, J.H.; visualization, T.N.; project administration, T.N.; funding acquisition, T.N. All authors have read and agreed to the published version of the manuscript.

**Funding:** The data collection was supported by a grant from the CLASS Excellence Fund from University of Idaho to Matthew Grindal. The analysis was supported by a grant from the Academic Senate, University of California at Riverside to Tanya Nieri.

**Institutional Review Board Statement:** The original data collection was approved by the Institutional Review Board of the University of Idaho (protocol #21-220, 20 December 2021). The secondary data analysis was deemed exempt by the Institutional Review Board of the University of California Riverside (protocol #22-056, 27 May 2022).

**Informed Consent Statement:** Informed consent was waived due to the study being identified as Exempt under Category 2 at 45 CFR 46.104(d)(2).

**Data Availability Statement:** Interested scholars should contact the authors for access to the data.

**Conflicts of Interest:** The authors declare no conflict of interest. The funders had no role in the design of the study; in the collection, analyses, or interpretation of data; in the writing of the manuscript; or in the decision to publish the results.

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
