# Peer review of "Contextual Factors in Ethnic-Racial Socialization in White Families in the United States"

_societies, doi:10.3390/soc13050114_

Round 1

Reviewer 1 Report

Here are some suggestions.

1.     Consider including a more diverse sample: The study focuses on white families, but it would be valuable to include families from different ethnic and racial backgrounds. This would provide a more comprehensive understanding of how different groups socialize their children around issues of race and ethnicity.

2.     Use a longitudinal study design: To gain a more complete understanding of how contextual factors influence ethnic-racial socialization, a longitudinal study design could be used to track families over time. This would allow researchers to observe how contextual factors change and how they impact ERS strategies over time.

3.     Consider broader societal factors: The study focuses on the influence of contextual factors such as neighborhood, school, and social network ethnic-racial composition on ethnic-racial socialization. However, it may be important to also consider broader societal factors, such as media representations and political discourse, that may also impact ERS strategies.

Author Response

We thank the reviewers for their reviews. Below, we respond to their reviews indicating how the paper was revised to incorporate their feedback.

Reviewer 1

Here are some suggestions.

  1. Consider including a more diverse sample: The study focuses on white families, but it would be valuable to include families from different ethnic and racial backgrounds. This would provide a more comprehensive understanding of how different groups socialize their children around issues of race and ethnicity.

The study focuses on U.S. white families by design. The majority of prior studies focus on families of color. Thus, our goal with this paper is to fill the gap in knowledge on white families. There do exist a few prior studies comparing racial-ethnic groups, but they tend to involve small, local samples. An advantage of our study is that it involves a large, national sample of white families.

  1. Use a longitudinal study design: To gain a more complete understanding of how contextual factors influence ethnic-racial socialization, a longitudinal study design could be used to track families over time. This would allow researchers to observe how contextual factors change and how they impact ERS strategies over time.

We agree that a longitudinal design would be ideal, and we acknowledge this point in the discussion, stating that future studies should be longitudinal. Again, a methodological strength of our study is the large, national sample. We view that as important progress in improving the rigor of existing empirical knowledge on this topic.

  1. Consider broader societal factors: The study focuses on the influence of contextual factors such as neighborhood, school, and social network ethnic-racial composition on ethnic-racial socialization. However, it may be important to also consider broader societal factors, such as media representations and political discourse, that may also impact ERS strategies.

We agree with the reviewer on this point. We state in the discussion that future research should include other factors. We have now added to that discussion the reviewer’s two mentioned suggestions.

Reviewer 2 Report

Abstract:  The first time you use ERS, please include a parenthetical explanation.  I assume it means ethnic racial socialization.  If so, in line 6-7 you could put (ERS) after the first use of the term, so it is clear next time.  This does appear at line 41 but should appear earlier.

line 15:  "experience preparation for" seems a bit awkward.  I think I understand what you mean.  Would it be better to say "have experiences that prepare them for bias..." ?

line 92:  preparation for bias -- does this imply they are prepared to be biased or that they are prepared to be aware that bias exists?  This seems to be addressed somewhat at 181-183 but would be helpful to review earlier in the essay.

Line 219:  Is the ) needed ?

Page 6:  Tabular Data:  This shows that 29.5% of the parents of those surveyed had attended college.  Is there any regional difference?  In the data supplied above, it seems that the incidence of college graduates in your surveyed population themselves was quite low (11-12%).  It seems this could be significant in some way if explored.  For instance, were racial attitudes or ERS different among those who attended college at an early age or were the children of people who did versus among those whose parents attended college as adults (and thus had passed their formative years)?  You address this somewhat at 286-288.  Were there regional differences in attitude by parental education levels?

Overall, I find this to be a very interesting, well-done, and powerful study that promises to engage future scholarship.  

General Comment

Is there any attempt in the literature to address what "whiteness" is?  I wonder if the current interest in ancestry and genealogy has had any effect on racial attitudes?  For instance, many "white" people who test discover they are ethnically 5% African or 4% Native American or a mixture of many different groups, including southern European, eastern European, and North African ancestry, as well as sub-Saharan African and/or Native American.  I wonder if this has had any effect on how "white people" think about themselves?

(Regarding this, I note that you do in your study identify your participants as themselves identifying as "white" and the children of "white" parents, so this may not be a particularly significant factor in your study, but it would be interesting to know if DNA/ethnicity tests have changed racial perceptions among whites and in what ways.)

As another General Comment:

Given that this is a recent (January 2022) survey and, as I understand the essay, one of the first of its kind, I wonder if there is any way to account for possible changes in racial attitudes that may have occurred during the pandemic?  Perhaps this was negligible, but is there any tendency, post-pandemic, for changes in racial attitudes in one direction or another?  Earlier, lines 22-30, you mention some of the larger political issues of the 2016-2020 period, but might the pandemic itself factor into this in any way?  You do mention (line 151) that the study occurred in the middle of a global pandemic, but there is no discussion about whether that might have had an significant effect on the findings.

General Comment regarding references:

My other experience reviewing for MDPI is usually that the author/publication date short reference is included in the text rather than reference numbers.  If that is needed, however, the editorial team usually works with the author to make necessary changes if a MS is accepted for publication, so I did not note this throughout my review.

Overall, I recommend accepting this article after minor revision.  It is soundly researched and produces significant conclusions.  I recommend that the author consider my comments above and make minor revisions where appropriate.

Author Response

We thank the reviewers for their reviews. Below, we respond to their reviews indicating how the paper was revised to incorporate their feedback.

Reviewer 2

Abstract:  The first time you use ERS, please include a parenthetical explanation.  I assume it means ethnic racial socialization.  If so, in line 6-7 you could put (ERS) after the first use of the term, so it is clear next time.  This does appear at line 41 but should appear earlier.

            We added the parenthetical explanation to the abstract.

line 15:  "experience preparation for" seems a bit awkward.  I think I understand what you mean.  Would it be better to say "have experiences that prepare them for bias..." ?

            We modified the language to remove the awkwardness.

line 92:  preparation for bias -- does this imply they are prepared to be biased or that they are prepared to be aware that bias exists?  This seems to be addressed somewhat at 181-183 but would be helpful to review earlier in the essay.

Preparation for bias involves messages about the possibility of becoming a victim of ethnic-racial discrimination. So, the strategy prepares the child for discrimination; it does not prepare the child to be biased. We added language in line 56 to clarify this point.

Line 219:  Is the ) needed ?

            We deleted the parenthesis.

Page 6:  Tabular Data:  This shows that 29.5% of the parents of those surveyed had attended college.  Is there any regional difference?  In the data supplied above, it seems that the incidence of college graduates in your surveyed population themselves was quite low (11-12%).  It seems this could be significant in some way if explored.  For instance, were racial attitudes or ERS different among those who attended college at an early age or were the children of people who did versus among those whose parents attended college as adults (and thus had passed their formative years)?  You address this somewhat at 286-288.  Were there regional differences in attitude by parental education levels?

As we state in the paper (Methods section), the rates of education in our sample of young adults are consistent with the national population of white young adults. We agree with the authors that one could examine whether education predicts different racial attitudes, but in this paper 1) we were interested in contextual predictors, not individual predictors and 2) we were interested in exposure to ERS strategies, not racial attitudes as outcomes of that ERS. Furthermore, our data do not include data on racial attitudes; thus, we cannot assess the topics recommended by the reviewer.

The number of college graduates in the sample is understandable given the age of the sample: 18-25 years. Some members of the sample may go on to complete college as they get older.

Overall, I find this to be a very interesting, well-done, and powerful study that promises to engage future scholarship.  

            Thank you.

General Comment

Is there any attempt in the literature to address what "whiteness" is?  I wonder if the current interest in ancestry and genealogy has had any effect on racial attitudes?  For instance, many "white" people who test discover they are ethnically 5% African or 4% Native American or a mixture of many different groups, including southern European, eastern European, and North African ancestry, as well as sub-Saharan African and/or Native American.  I wonder if this has had any effect on how "white people" think about themselves?

There is a literature on whiteness and white identity. However, our study focuses on process, not outcomes. We focus on factors (A) that relate to the process of socialization (B). We do not focus on outcomes of socialization – the actual identities and attitudes (C). We have other research in development that does that, but our goal in this paper was to better understand how contextual factors (A) shape the socialization to which white children are exposed - that is, the process of socialization (B).

 (Regarding this, I note that you do in your study identify your participants as themselves identifying as "white" and the children of "white" parents, so this may not be a particularly significant factor in your study, but it would be interesting to know if DNA/ethnicity tests have changed racial perceptions among whites and in what ways.)

There is a small but emerging literature on the social impacts of commercial ancestry discovery products (e.g., 23 and Me, ancestry.com). However, genetics/DNA and/or knowledge/learning about those things are individual factors, and we were interested in contextual factors. Hence, these variables are beyond the scope of our paper.

As another General Comment:

Given that this is a recent (January 2022) survey and, as I understand the essay, one of the first of its kind, I wonder if there is any way to account for possible changes in racial attitudes that may have occurred during the pandemic?  Perhaps this was negligible, but is there any tendency, post-pandemic, for changes in racial attitudes in one direction or another?  Earlier, lines 22-30, you mention some of the larger political issues of the 2016-2020 period, but might the pandemic itself factor into this in any way?  You do mention (line 151) that the study occurred in the middle of a global pandemic, but there is no discussion about whether that might have had an significant effect on the findings.

Participants reported on messages received in childhood, not in the current period. To capture pandemic effects we would have to gather data on a younger sample. We added a suggestion for future research on this topic in the discussion.

We mention the current events 2016-2020 to justify an exclusive focus on white families, since they were relatively unexamined in the prior period.

General Comment regarding references:

My other experience reviewing for MDPI is usually that the author/publication date short reference is included in the text rather than reference numbers.  If that is needed, however, the editorial team usually works with the author to make necessary changes if a MS is accepted for publication, so I did not note this throughout my review.

We followed this particular journal’s format for in-text citations.

Overall, I recommend accepting this article after minor revision.  It is soundly researched and produces significant conclusions.  I recommend that the author consider my comments above and make minor revisions where appropriate.

            Thank you.